# Clinical Depression and Anxiety Are Relieved by Microvascular Decompression in Patients with Trigeminal Neuralgia—A Prospective Patient-Reported Outcome Study

**DOI:** 10.3390/jcm13082329

**Published:** 2024-04-17

**Authors:** Marlies Bauer, Aleksandrs Krigers, Victoria Schoen, Claudius Thomé, Christian F. Freyschlag

**Affiliations:** Department of Neurosurgery, Medical University of Innsbruck, Anichstraße 35, 6020 Innsbruck, Austria; marlies.bauer@tirol-kliniken.at (M.B.);

**Keywords:** trigeminal neuralgia, microvascular decompression, depression, surgical outcome, posterior fossa surgery

## Abstract

**Objective:** Patients with idiopathic trigeminal neuralgia (TN) live in constant fear of triggering shock-like pain episodes, which may cause symptoms of depression and a reduction in quality of life. Microvascular decompressive surgery has been demonstrated to achieve satisfactory and stable results. With this study, we wanted to investigate prevalence and risk factors for depression and perceived stress in correlation with symptom relief after surgical treatment. **Methods:** In this prospective study, patients undergoing microvascular decompression (MVD) for TN were included. The Barrow Neurological Institute Pain Score (BNI), Beck Depression Inventory (BDI), Chronic Pain Acceptance Questionnaire (CPAQ), Perceived Stress Questionnaire (PSQ) and McGill questionnaire were used to evaluate depression, stress and anxiety disorders before and 3 months after MVD. **Results:** A total of 35 patients (16 male (46%)) with a mean age of 55.4 (SD 15) years were included in this study. The BDI revealed that 24 (68.8%) patients harbored mild-to-extreme depression preoperatively (2.4 ± 1.4), which improved to 1.2 (±0.6, *p* < 0.0001). Pain acceptance also changed from 64 (±11.3) to 67.7 (±9.3, *p* = 0.006). Perceived stress decreased from 46.9 (±21.9) to 19.6 (±18.6) (*p* < 0.0001) postoperatively, and pain decreased from 31.0 (±11.7) to 9.4 (±12.9, *p* < 0.0001). Microvascular decompression reduced the mean BNI pain score significantly from 4.6 to 1.8 postoperatively (*p* < 0.00001). **Conclusions:** Depression and perceived stress are prevalent in patients with idiopathic TN. Adequate treatment not only provides a high rate of satisfaction through pain relief, but also leads to immediate and significant improvements in depression and stress. Thus, in patients with TN who do not reach an adequate and timely pain reduction through medical management and develop signs of depression, early treatment with microvascular decompression should be considered.

## 1. Introduction

Trigeminal neuralgia (TN) is a devastating neuropathic disorder marked by excruciating episodic, electrical shock-like pain along one or more branches of the trigeminal nerve. Often triggered by seemingly innocuous sensory stimuli, such as washing the face or chewing, TN can severely impact patients’ quality of life [1]. Notably, approximately half of TN patients also report experiencing a persistent and continuous lower-intensity pain in the affected area, adding to the disease’s burden [2]. The standard of care involves a stepwise approach, starting with the administration of antiepileptic drugs like carbamazepine and oxcarbazepine, which are the cornerstone of medical management according to current clinical guidelines [3]. Despite their widespread use, a significant proportion of patients either do not achieve complete relief or cannot tolerate the side effects of these medications [4], leading to the consideration of surgical options. 

Beyond medical and surgical management, strategies for TN may also include interventional procedures such as percutaneous techniques (e.g., balloon compression, glycerol rhizotomy and radiofrequency thermal lesioning) and non-invasive modalities like stereotactic radiosurgery [2,3,4,5,6]. Each treatment option carries its own risk–benefit profile and potential impact on the patient’s quality of life [7,8,9], highlighting the importance of personalized treatment planning. 

The role of early microvascular decompression (MVD) remains a subject of debate, and current guidelines do not support early MVD prior to medical management [3]. However, there is a subset of studies indicating that patients would prefer early surgical treatment over medical management when thoroughly informed with data regarding expected outcomes, complications and adverse events [10]. With a long-term success of approximately 80% and a low recurrence rate [11,12], microvascular decompression constitutes an effective and safe treatment. However, it is crucial to underscore that no treatment guarantees a complete remission of pain [6,13], so patients are persistently anxious that they may encounter recurrent attacks [14]. This persistent anxiety underscores the complexity of managing TN. 

The implications of TN extend far beyond physical pain, significantly affecting patients’ social lives and psychological well-being. Chronic pain and the ongoing need for medication are closely associated with the development of depression and anxiety, leading to a marked decline in quality of life (QoL) [15,16]. Patients experiencing depression and anxiety with consequently reduced QoL show a worse prognosis compared to patients harboring only pain [17]. This interplay between physical pain and mental health underscores the necessity of a comprehensive approach to treating TN, one that addresses both facets of the disease. The intersection of chronic pain and mental health disorders is a complex and underexplored domain within TN research. Prospective studies, which can offer insights into the causality and progression of depression and anxiety in the context of TN, are notably scarce. This gap in the literature highlights a critical need for research designed to longitudinally assess the impact of TN on mental health outcomes. Therefore, our study aims to fill this gap by investigating the prevalence and progression of depression, perceived stress and pain in a prospective cohort of patients undergoing MVD for TN. By adopting this prospective study, we seek to elucidate the relationship between surgical intervention and changes in mental health status, offering valuable insights into the potential benefits of MVD beyond pain relief. We hypothesize that MVD, by effectively managing pain, will significantly reduce symptoms of depression and anxiety, leading to an improvement in patients’ overall quality of life. While this study comprehensively investigates the prevalence and progression of depression and perceived stress in this patient cohort, it is important to note that anxiety, although closely related and inherently significant to psychological impact, was not measured as a separate outcome in this research. Overall, this approach not only aligns with the growing recognition of the biopsychosocial model of chronic pain management, but also underscores the importance of integrating mental health care into the treatment paradigm of TN. Such insights are vital for informing clinical practice, guiding treatment decisions and improving patient counseling and consent processes. Furthermore, by shedding light on the psychological ramifications of trigeminal neuralgia and the potential for surgical interventions to alleviate not only physical pain but also mental distress, our research contributes to a more holistic understanding of TN and its treatment. 

## 2. Material and Methods

We conducted a prospective longitudinal study of patients undergoing MVD for classical and idiopathic TN. Exclusion criteria included patients with secondary causes of TN or clinical evidence of multiple sclerosis. This determination was based on a comprehensive review of medical histories, physical examinations and diagnostic imaging, including MRI scans.

In accordance with established diagnostic guidelines, all patients included in the study were diagnosed using the International Classification of Headache Disorders, 3rd edition (ICHD-3), and the International Consensus on Pain (ICOP) criteria. These criteria were rigorously applied by experienced clinicians to ensure the accurate classification of patients.

The study was approved by the local ethics committee (AN2016-0074, 361/4.7) and is reported in accordance with the STROBE checklist for observational studies [18]. 

All patients completed an assessment set of multiple questionnaires. Demographic data including sociodemographic characteristics (age, gender, family and educational status, sport activities, weight, alcohol intake and smoking habits) and clinical data (duration of symptoms, medication, side of pain, divisions of the trigeminal branches affected) were recorded (Table 1). Patients were assessed with questionnaires at two timepoints: before surgery and 3 months after surgery. To evaluate the impact of trigeminal neuralgia and its treatment on patients, we utilized several validated assessment tools. Each tool was selected based on its reliability and validity in measuring specific aspects of TN, including pain severity, chronic pain acceptance, depression and perceived stress.

Pain was graded using the 5-point Barrow Neurological Institute Pain Score (BNI), where 1 stands for “no pain, no medication” and 5 represents “severe pain—no pain relief” [19]. The BNI score is a widely recognized scale for assessing pain in TN patients, with proven reliability and validity.

The severity of pain was evaluated using the McGill Pain Questionnaire (MPQ). The MPQ is a multidimensional pain assessment scale, consisting of three parts measuring dimensions of pain experience (sensory-discriminative, affective-motivational and cognitive-evaluative) [20]. It can be used to evaluate a person experiencing significant pain and to determine the effectiveness of any intervention. It includes three sections: “What does your pain feel like?”, “How does your pain change with time?” and “How strong is your pain?”.

Acceptance of chronic pain was measured using the Chronic Pain Acceptance Questionnaire (CPAQ), which is a 20-item questionnaire with 2 subscales (pain willingness and activities engagement). The acceptance of chronic pain is thought to reduce unsuccessful attempts to avoid or control pain and thus focus on engaging in valued activities and pursuing meaningful goals [21].

The Beck Depression Inventory (BDI) is a common instrument to assess the presence and severity of depression. It has been extensively studied, showing a high test–retest reliability across various populations [22]. It comprises 21 items and each item is evaluated on a 4-point scale (from 0 to 3), yielding a total BDI score ranging from 0 to 63. A higher total score indicates increased severity of depression (Table 2). A total BDI score > 10 reflects a clinically significant mood disturbance reaching up to a BDI > 40, reflecting extreme depression.

The Perceived Stress Questionnaire (PSQ-20) was used to objectively measure the amount of stress patients experience. The perceived stress scale provides four subscales—“worries”, “tension” and “joy” as stress responses, and “demands” as the perception of external stressors—and thereby emphasizes the subjective experience of stress [23,24,25,26]. Higher scores indicate a higher severity of perceived stress. The resulting total score is transformed between zero and one (linearly, PSQ = (raw value − 30)/90).

All questionnaires were administered in their approved German language versions, ensuring linguistic and cultural appropriateness for our patient population. The translation and validation processes for these versions adhered to international standards, ensuring the reliability and validity of the tools remained intact.

### Statistical Analysis

SPSS was used for statistical analyses (IBM SPSS Statistics for Mac, Version 26.0. IBM Corp.: Armonk, NY, USA). The mean is expressed as ± SD throughout (for normal distributed values) or with the median and interquartile range. Descriptive statistics were used to summarize patient characteristics. The normality assumption was evaluated using the Kolmogorov–Smirnov test. The Wilcoxon test was used to compare pre- and postoperative scores. Pearson’s or Spearman’s correlation coefficients (r) were used to test the association of two continuous measures. 

The results and scores before and after MVD treatment were compared using Student’s *t*-test. Associations were considered statistically significant when *p* < 0.05. 

## 3. Results

### 3.1. Participants

A total of 35 patients were enrolled in this study. The demographical data of all patients are shown in Table 1. The mean age of patients (16 (46%) males) was 55.4 (SD ± 15). The median time from initial diagnosis (symptoms) until surgical decompression was 5 years (IQR: 2–8 years). The distributions and combinations of affected trigeminal roots were as expected: V1 was solely affected in 2.9% (*n* = 1), V2 in 17.1% (*n* = 6) and V3 in 22.9% (*n* = 8) (see Table 1). Two patients (5.7%) had an affection of V1 + V2, fourteen patients (40%) had V2 + V3 involved, and four patients (11.4%) suffered pain in all three branches of the trigeminal nerve (V1 + V2 + V3). The age of the patients showed no influence on any of the investigated scores or outcome variables; however, older patients showed a higher number of involved trigeminal divisions (r = 0.424, *p* = 0.011). 

### 3.2. Patient-Reported Outcomes after MVD

The median preoperative BNI pain score of 5 (IQR: 5–5) was significantly reduced to a median of 1 (IQR: 1–3) postoperatively (*p* < 0.001), showing that patients had a significant pain reduction after surgery. Furthermore, patients with a BNI score > 1 demonstrated a highly significant reduction in pain medication needed (*p* < 0.0005). 

### 3.3. McGill Pain Questionnaire

The results of the McGill Pain questionnaire were in accordance with the BNI scores. Preoperatively, the median McGill score was 32 (IQR 28–40), which decreased to 0 (IQR 0–19) after surgery. Median changes in the McGill pain questionnaire were −25 (IQR −33–−14). An insufficient reduction in the McGill score was noted in patients with regular alcohol consumption (*p* = 0.021).

### 3.4. Depression

Preoperatively, 68.6% of patients (*n* = 24) harbored a clinically relevant mood disturbance with a mean BDI of 15.7 points. Postoperatively, the patients significantly reduced in BDI score with a mean of 5.8 points, representing no clinical depression. Postoperative sport activities showed a beneficial influence on postoperative BDI (*p* = 0.04)

### 3.5. Pain Acceptance

The Chronic Pain Acceptance Questionnaire (CPAQ) showed a generally high pain acceptance with a mean preoperative pain acceptance score of 64.06 (SE 1.92) and a postoperative score of 67.69 (SE 1.57). The comparison of pre- and postoperative CPAQ results revealed significant changes after surgical intervention (*p* = 0.006). Further, the CPAQ results were closely related to personal habits and coping mechanisms to overcome pain-related restrictions. The CPAQ is related to sports activities, and patients who preoperatively enjoyed sports activities demonstrated significantly higher pain acceptance (+8 points, SE+/−4, *p* = 0.049, R2 = 0.112). Patients with affection of the trigeminal root V2 showed decreased pain acceptance during follow-up (*p* = 0.09). Additionally, patients with frequent preoperative alcohol intake were characterized by a significant decrease in postoperative pain acceptance (*p* = 0.02). Patients who continued to consume alcohol regularly showed a lesser reduction in the McGill Pain Questionnaire (*p* = 0.021), indicating a high likelihood of postoperative residual pain that did not influence the BNI pain score.

### 3.6. Perceived Stress

Significant differences between pre- and postoperative perceived stress levels were present. The median Perceived Stress Questionnaire result amounted to 46.70 (IQR 31.67–66.67) preoperatively and decreased after surgery to a median of 13.33 (IQR 5–26.07). In accordance with other scores, the changes in PSQ results (less stress, i.e., a decrease) after surgery were lower if patients regularly consumed alcohol (*p* = 0.049), but higher if the patients were actively involved in sports activities (*p* = 0.045).

### 3.7. Bodyweight

The PSQ results correlated with the patients’ weight (preop: r = 0.372, *p* = 0.028; postop: r = 0.348, *p* = 0.041), indicating that patients with TN and higher bodyweight suffered higher levels of perceived stress. Weight also negatively influenced the BNI score significantly at both timepoints (preop: r = 0.385, *p* = 0.022; postop: r = 0.453, *p* = 0.006). However, postoperative depression was more frequent in patients with lower bodyweight (r = −0.368, *p* = 0.03).

Gender-specific changes could only be identified in the depression inventory: preoperatively, female patients showed a higher score in the BDI, reflecting a higher degree of depression (*p* = 0.047), whereas male patients showed a significantly higher change from pre- to postoperative BDI score (*p* = 0.048). All other scores failed to show gender-related differences. The chronic pain acceptance delta was higher postoperatively in female patients with +7 (SE +/−3, *p* = 0.04, R^2^ = 0.122)

Generally, our investigation revealed significant postoperative changes in lifestyle habits: it was demonstrated that patients significantly reduced their smoking habits (*p* = 0.046) and increased sports activities (*p* = 0.034) after surgical treatment. 

Surgically induced complications occurred in three patients (two cerebrospinal fluid (CSF) leaks and one mild transient facial palsy). 

## 4. Discussion

In this study, we analyzed the prevalence of depression and pain-related distress in 35 consecutive patients with idiopathic TN who underwent MVD in our department. While our results align with the existing literature indicating a higher risk of mental health disorders in patients suffering from chronic pain, including TN, the variability in reported prevalence rates demands further examination [8,27,28]. The psychological impact of TN extends beyond mere pain sensation: it encompasses a spectrum of pain-related distress, social isolation and impaired daily functioning, all of which can exacerbate symptoms of depression and anxiety. According to our results, we show that 24 out of 35 patients suffered from at least mild depression preoperatively. The intermittent and unpredictable nature of TN pain contributes to a constant state of apprehension and distress, further isolating patients from special support systems and hindering their ability to engage in daily activities. This cycle of pain and psychological distress underscores the need for a holistic approach to treat both the physical and mental health of those affected. 

Depression is a widespread mental health issue in Europe, affecting approximately 7% of the population according to the WHO’s European Health Information Gateway (2017). Depression is notably more prevalent in patients suffering from chronic pain. A comprehensive systematic review by Meda et al. from 2022 reported a significantly higher prevalence of depression among patients with chronic pain, ranging from 13% to 85% based on the type of chronic pain condition [29]. The large variability in prevalence rates can be attributed to the diverse types of chronic pain conditions and the methodologies used to assess depression. This emphasizes the crucial need for effective mental health support and integrated care strategies for individuals suffering from chronic pain. Our study demonstrated the prevalence of depression in patients with TN to be 68.6%.

A recent study on TN showed that one-third of patients had mild-to-severe depression and over 50% were anxious [28]. An epidemiological study revealed increased anxiety, depression and insomnia in patients with TN, highlighting its effect on mental health [30]. 

Being a stressful and painful condition, TN pain plays a significant role in incapacitating the individual and places a substantial burden on quality of life, making it difficult to perform the physical or mental activities that could previously be carried out normally. However, the psychological consequences of TN, especially depression and anxiety, have seldom been highlighted and are likely to be underdiagnosed. This can probably be attributed to the common consideration that facial pain is intermittent, non-life-threatening and can be effectively treated. Microvascular decompression, by offering potential for significant pain relief, plays a crucial role in interrupting this cycle. The reduction in pain-related distress through successful decompression can restore a patient’s functional capacity and enhance their overall quality of life.

Overall, MVD is widely regarded as a safe and effective treatment for TN, with a high long-term success rate. In the present study, 21 patients (60%) achieved complete pain relief without medication (BNI pain score I), 12 patients (34.3%) achieved partial pain relief (BNI pain score II–III), and only 2 patients (5.7%) failed to obtain pain relief (BNI pain score IV–V) at the first postoperative visit. In addition, most of the postoperative complications were reversible. Our findings suggest that the alleviation of physical pain is closely linked to improvements in mental health, with patients experiencing notable reductions in depression post-surgery. This relationship highlights the potential of microvascular decompression not only as a physical intervention but as a means of psychological rehabilitation. 

However, translating these findings into clinical practice involves navigating a complex landscape of challenges and barriers. Resource limitations, patient preferences and varying levels of clinician expertise can all impact the decision-making process and ultimately affect patient outcomes. For instance, based on a low level of evidence, medical management is recommended before offering surgery for TN [5,31,32,33]. Therefore, for the majority of cases, surgical intervention is reserved for those with a longer symptomatic period or patients who develop significant side effects early on [34]. Even though two-thirds of patients during conservative treatment report to suffer from disabling side effects accompanied by decreasing efficacy over time, the time to interventional treatment can be long [3]. 

Furthermore, since the efficacy of medical management typically decreases over time, many patients turn to surgical intervention for more durable pain relief [7]. In addition, the scientific evidence for the drug treatment of TN is weak [19].

In addressing these challenges, it becomes clear that the effective management of trigeminal neuralgia, particularly in cases where mental health is significantly impacted, requires a multidisciplinary approach. Incorporating mental health support and counseling into TN treatment protocols can help prepare patients for the journey ahead, ensuring they are fully informed of their options and the potential outcomes of each. 

By providing effective pain relief, microvascular decompression has the potential to break this cycle, reducing pain-related distress and thereby alleviating symptoms of depression. The mechanisms through which MVD achieves these psychological benefits likely involve both the direct alleviation of the physical trigger of distress (i.e., the pain itself) and the secondary effects of pain relief on psychological well-being, such as improved sleep, increased ability to engage in social and physical activities, and a restored sense of control over one’s life. Our findings, showing significant improvements in depression and stress symptoms post-MVD, underscore the importance of early surgical intervention in the treatment algorithm for trigeminal neuralgia, especially for patients with a significant mental health burden. The results of the present study indicate that the depression and stress symptoms in TN patients were significantly improved after MVD. We compared the mean BDI, CPAQ and PSQ scores before and weeks after surgery, showing that all of these scores decreased significantly. 

As shown in a previous study, these scores were significantly improved in MVD patients, whereas medically treated patients showed a significant improvement only in anxiety scores [35].

In conclusion, our study contributes to the growing body of evidence highlighting the significant mental health burden associated with TN and the potential of surgical interventions like MVD to alleviate this burden. By shedding light on the complex interplay between physical pain and psychological distress, we underscore the need for treatment approaches that address the multifaceted nature of this debilitating condition. As we move forward, the challenge lies in integrating these insights into clinical practice, ensuring that patients with TN receive comprehensive care that addresses both their physical and mental health needs, ultimately improving their quality of life and offering hope amidst the pain.

As such, the early consideration of microvascular decompression in the treatment algorithm, especially for patients exhibiting signs of clinical depression or severe anxiety, could significantly enhance the quality of life and psychological well-being of those afflicted by this condition.

## 5. Limitations

While our study highlights the promising potential of early microvascular decompression in managing trigeminal neuralgia, several limitations should be carefully considered. Firstly, the relatively small sample size, while sufficient for initial observations, limits the generalizability of our findings. The complexity and variability inherent in TN cases necessitate larger cohort studies to more accurately assess the wide range of patient responses to MVD. Additionally, the follow-up period in our study was relatively short. Given the chronic nature of TN and the potential for long-term complications and the recurrence of symptoms, longer follow-up periods are crucial for understanding the durability of MVD benefits over time. Such data would be invaluable in guiding patient counseling and decision-making processes regarding treatment options.

Another limitation is the lack of a control group for comparison, such as patients receiving medical management alone or undergoing alternative surgical and non-surgical interventions. This comparison could provide more definitive insights into the relative efficacy of MVD versus other treatment modalities, addressing the critical question of how best to manage TN to optimize patient outcomes. 

Moreover, our study did not account for potential confounding factors such as patients’ previous treatments, mental health status before developing TN and other comorbid conditions, which could influence both the impact of TN on quality of life and the outcomes of MVD. A more detailed analysis including these factors would contribute to a more nuanced understanding of MVD’s role in TN management. 

Finally, while our study provides important insights into the impact of MVD on depression and perceived stress in TN patients, the subjective nature of these measures and the reliance on self-reported questionnaires underscore the need for incorporating objective biomarkers for stress and mental health in future research. This could enhance the robustness of findings related to the psychological benefits of MVD.

In conclusion, while our study suggests the potential benefit of the early consideration of microvascular decompression in certain cases of trigeminal neuralgia, these limitations underscore the need for caution in drawing definitive conclusions. Further research, involving larger cohorts, extended follow-up periods and objective measures is essential to validate these findings and deepen our understanding of the complexity of trigeminal neuralgia.

## 6. Conclusions

Depressive symptoms are highly prevalent in patients with trigeminal neuralgia. Therefore, we recommend the assessment of depression and anxiety in patients at diagnosis and periodically during the further medical management of TN. Microvascular decompression not only provides high rates of satisfaction through pain relief, but simultaneously leads to significant improvements in depression and stress scales. There is a strong argument for physicians to encourage patient referrals to neurosurgical evaluation earlier in the course of the disease.

## Figures and Tables

**Table 1 jcm-13-02329-t001:** Patient characteristics.

	*n* (%)
	*n* = 35
Age, mean (SD)	55 (15)
Sex	
Men	16 (46%)
Women	19 (54%)
Family Status	
Married/In a Relationship	24 (69%)
Single/Divorced/Widowed	11 (31%)
Children	
Yes	25 (72%)
No	10 (28%)
Education Status	
Primary School	3 (9%)
Middle/High School	26 (74%)
University	6 (17%)
Smoking	9 (26%)
Regular Alcohol Consumption	8 (23%)
Division Affected	
V1	1 (2.9%)
V2	6 (17.1%)
V3	8 (22.9%)
V1 + V2	2 (5.7%)
V2 + V3	14 (40%)
V1 + V2 + V3	4 (11.4%)
Side of Pain	
right	20 (57%)
left	15 (43%)

**Table 2 jcm-13-02329-t002:** Monovariate analysis of pre- and postoperative results. All but CPAQ values given as median with IQR. CPAQ follows a normal distribution; therefore, mean values with SE are given.

	Preop	IQR/SE	Postop	IQR/SE	DELTA	IQR/SE	*p*
BNI	5	(5–5)	1	(1–3)	−3	(−4–−2)	
BDI	2.0	(1–4)	1.0	(1–1)	−2		<0.001
CPAQ	64.06	(SE 1.92)	67.69	(SE 1.57)	3.63	(SE 1.75)	<0.001
MPQ	32	(28–40)	0.0	(0–19)	−25	(−33–−14)	<0.001
PSQ-20	46.7	(31.7–66.7)	13.3	(5.0–26.7)	−28.33	(−51.67–−13.33)	<0.001

(BNI = Barrow Neurological Institute Pain Score (1 = no pain, no medication; 5 = severe pain), BDI = Beck Depression Inventory, CPAQ = Chronic Pain Acceptance Questionnaire (higher value = better acceptance), MPQ = McGill Pain Questionnaire (lower value = less pain), PSQ-20 = Perceived Stress Questionnaire (higher values = more perceived stress).

## Data Availability

The data presented in this study are available on request from the corresponding author due to legal reasons.

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
