# Peer review of "Clinical Depression and Anxiety Are Relieved by Microvascular Decompression in Patients with Trigeminal Neuralgia—A Prospective Patient-Reported Outcome Study"

_jcm, 2024, doi:10.3390/jcm13082329_

Round 1
Reviewer 1 Report
Comments and Suggestions for Authors
In introduction
Some statements lack citations to support claims, such as the assertion that "early microvascular decompression in TN is not supported by strong evidence."
It does not clearly articulate why it is important to investigate the prevalence of depression, anxiety, perceived stress, and pain in a prospective cohort of patients undergoing microvascular decompression (MVD) for TN.
The introduction briefly mentions the impact of chronic pain and medication on depression and anxiety but does not delve into the broader context of mental health issues in TN patients.
The introduction mentions that the mental consequences of TN have not been analyzed prospectively but does not elaborate on why a prospective study design is specifically chosen for this investigation.
It does not clearly define the specific research questions or hypotheses guiding the study.
Some statements in the introduction, such as “all known treatment options fail to guarantee complete remission of pain,” may be overly generalized without providing specific evidence or qualifications.
In methodology
It mentions exclusion criteria for patients with secondary causes of TN or clinical evidence of multiple sclerosis, it does not elaborate on how these criteria were determined or verified.
It would be beneficial to provide more information on the reliability and validity of these assessment tools, including any validation studies conducted in similar patient populations.
The methodology mentions the use of approved German language versions of all questionnaires. While this is important for ensuring the accuracy of the assessments in the study population, it would be helpful to provide information on how the translations were validated and whether any cultural adaptations were made.
In Discussion
The discussion correctly acknowledges the variability in reported prevalence rates of depression among patients with chronic pain conditions, including TN. However, it would be helpful to further discuss potential factors contributing to this variability, such as differences in study populations, assessment tools, and cultural factors, to provide a more nuanced understanding of the findings.
It could benefit from further elaboration on the specific psychological mechanisms underlying this impact, such as the role of pain-related distress, social isolation, and impaired functioning in exacerbating depressive and anxiety symptoms in TN patients.
It would be valuable to discuss potential mechanisms through which MVD may alleviate depression and anxiety symptoms, such as reducing pain-related distress, restoring functional capacity, and enhancing overall quality of life.
Discuss potential challenges or barriers to implementing these recommendations in real-world clinical settings, such as resource limitations, patient preferences, and clinician expertise.
Reviewer 2 Report
Comments and Suggestions for Authors
The Authors present a very interesting and useful prospective study, demonstrating that patients affected by TN chronic pain are at a higher risk of developing depression and anxiety than the general population. Therefore, Chronic pain due to idiopathic TN has to be treated at an early stage to preserve mental health. These results are usefully encouraging to show that Microvascular Decompression either provides highest rates of satisfaction through pain-relief or leads to significant improvements in depression and stress scales. Finally, a good counselling to patients could be offered by Clinicians to refer them to neurosurgical treatment in the early stage of the disease, improving the whole management of this pathology, granting a high quality of life to patients.
Reviewer 3 Report
Comments and Suggestions for Authors
Dear Sir,
I have the following comments regarding your article.
The aim of the study is really important. I have the following comments to offer
Abstract
24 with a percentage may be mentioned.
Is there a difference in anxiety and depression in the patients who had completed relieved of pain verus those who did not?
It may be more appropriate to also analyze those with >50% reduction vs less than that, to feel the real effect.
The last sentence of the abstract is not based on the author's study as they do not compare these parameters between medical and surgical groups.
Methods
Have you excluded TN with the vascular loop? If not, how many had vascular loops? Is there any difference in response between them?
What pain severity were your inclusion criteria?
German may be German
Have you considered 2- sided p-value?
The categorization of the patients (as commented in the abstract section) and the statistical analysis need elaboration
Sample size calculation is missing.
Results
Over and above the raw score comparison, the improvement by >50% of not may be also compared.
How many had total remission?
Discussion
Authors may discuss medication response and how their surgery is superior to drugs.
TN-related disability may be more focused than pain as a whole.
Conclusion
Is too assertive, needs to modify based on their results
Round 2
Reviewer 1 Report
Comments and Suggestions for Authors
All concerns addressed successfully
thank you for the acknowledgment
I see this paper fly to new heights soon
congratulations to the authors.
Reviewer 3 Report
Comments and Suggestions for Authors I have gone through the paper. The comments are as follows: The author has addressed most of the comments. However, the anxiety and depression scores are not mentioned separately.Author Response
Please see the attachment.
